# Engaging in Sustainable Consumption: Exploring the Influence of Environmental Attitudes, Values, Personal Norms, and Perceived Responsibility

**Aistė Čapienė [1],\*** , **Aušra Rūtelionė [1],\*** and **Krzysztof Krukowski [2]**

1 Faculty of Bioeconomy Development, Vytautas Magnus University, LT-44248 Kaunas, Lithuania
2 Institute of Management and Quality Sciences, University of Warmia and Mazury in Olsztyn, 10-719 Olsztyn, Poland
\* Correspondence: aiste.capiene@vdu.lt (A.Č.); ausra.rutelione@vdu.lt (A.R.)

**Abstract:** This study explores the links between environmental attitudes and values, personal norms, perceived responsibility, pro-environmental and prosocial engagement in sustainable consumption, and sustainable consumption behavior. Data was collected by surveying 904 Lithuanians through non-random quota sampling. Empirical research reveals that internal factors, such as environmental attitudes, values, personal norms, and perceived responsibility, have a positive direct effect on engagement with sustainable consumption. In addition, the findings indicate that pro-environmental and prosocial engagement to act as a mediator in enhancing the impact on sustainable consumer behavior. The results of this study expand the understanding of the engagement phenomena and how it can assist in shifting to sustainable consumer behavior in the Lithuanian context. Opportunities to encourage sustainable consumption behavior are presented for marketers and policy makers.

**Keywords:** environmental attitudes; environmental knowledge; personal norms; perceived responsibility; pro-environmental and prosocial engagement in sustainable consumption and sustainable consumption behavior

## 1. Introduction

The continuous rise in demand for goods and services and the habitualization of consumption practices have resulted in the growth of an excessive consumption culture. The prevalence of materialistic lifestyles (initially exclusive to developed countries), the growth of the global population, and the rising wealth of both high- and low-income groups [1] provide excellent conditions for this. Overconsumption has had a massive impact on the environment and communities worldwide, exacerbating global challenges, such as climate change, biodiversity loss, and environmental degradation [2]. Sustainable consumption behavior is often described as responsible, ecological, or socially friendly [3]. It is also associated with changes in consumer behavior [4], since it implies buying exclusive (organic, green, or fair trade) products or consuming less [5,6]. Sustainable consumption behavior can be described through the following three dimensions: quality of life, care for environmental well-being, and care for future generations [7] It could reduce risks to human health and the environment [8], with consumers playing a key role in facilitating social change [5]. Furthermore, Orîndaru et al. [9] noted that, during the COVID-19 pandemic, consumers have started to change their habits and purchasing decisions, including those determined within discounted products as well as local and fresh product buying.

The phenomenon of consumer engagement, first studied in the twenty-first century, relates to consumer readiness to actively participate and communicate with the object [10–12]. This phenomenon has only recently been analyzed, and the main research underlying the theoretical assumptions of this work over the past five years is a relatively new direction in research. However, the results of empirical research confirming the positive consequences

of pro-environmental and prosocial engagement are consistent—the more consumers engage in sustainable consumption, the more it is certain their sustainable consumption behavior will change [12]. Pro-environmental and prosocial consumer engagement in sustainable consumption, which can be described as participating and supporting related events, communication, and involving others and maintaining a conscious focus on environmental and social issues, leads to behavioral change, thus, contributing in the struggle to solve global issues [13]. Such engagement has a stronger effect than pro-environmental or prosocial behavior, as it integrates consumer communities united by a conscious focus on environmental and social issues [13]. Meanwhile, engagement, which is described in this study as a consumer's psychological state that expresses a desire to act in the interest of the environment and public interest, is a multidimensional concept consisting of the following three dimensions: conscious attention, enthusiastic participation, and social connection [10,13].

The topic of engagement in sustainable consumption has caused a great interest among researchers, but research on the links between pro-environmental and prosocial engagement and sustainable consumption behavior is still sparse. Studies on sustainable consumption identify consumer engagement as a factor in promoting sustainable consumption behavior [14–18]. It should be noted that some studies analyze the direct relationship between engagement and sustainable consumption behavior [19,20], while others approach engagement as a mediating factor between determinants of engagement and sustainable consumption behavior [12,13]. Some studies combine pro-environmental and prosocial aspects of engagement [10,13], while others choose to focus either on the prosocial [21,22] or pro-environmental [23] aspect.

Research papers [11–13,24,25] allow us to hypothesize positive associations between pro-environmental and prosocial engagement and sustainable consumption behavior, and these papers highlight the need for further research. Recent research focuses on exploring factors that determine consumers' pro-environmental and prosocial engagement in sustainable consumption [12,13]. Researchers have also developed some alternative approaches to the factors causing consumers' pro-environmental and prosocial engagement, which mostly depend on the variety of engagement objects and contexts.

Adopting the stimulus–organism–response (SOR) paradigm allows researchers to present pro-environmental and prosocial engagement as an organism influenced by different stimuli. The organism is usually defined as the psychological state, as pro-environmental and prosocial engagement are defined through the psychological state of an individual that can cause different consumer reactions (effects) [25,26]. In this study, the expected response is sustainable consumption behavior that is conceptualized as meeting basic human needs, ensuring quality of life and material well-being, reducing use of resources, waste, and pollution throughout the product or service life cycle, and concern for future generations [7]. Only a few researchers have addressed the question of pro-environmental and prosocial engagement as a mediator between stimuli and response [12]. Factors that influence consumers' pro-environmental and prosocial engagement can be treated as internal or external stimuli. Some studies indicate that internal factors have a bigger effect on engagement and behavior [11,12]. In our research, we aim to explore the effect of four internal factors, namely environmental values [27], environmental attitudes [10,13], personal norms [28], and perceived responsibility [18,29], on pro-environmental and prosocial engagement. At the same time, we also seek to reveal the role of consumers' pro-environmental and prosocial engagement as a mediator between stimuli and response, i.e., sustainable consumption behavior. Thus, our key questions are as follows: *How do consumers' environmental attitudes, environmental values, personal norms, and perceived responsibility influence pro-environmental and prosocial engagement in sustainable consumption, and how does this engagement in turn relate to sustainable consumption behavior?* In addition, we contribute to the body of sustainable consumption research while widening the geographical scope through introducing results from a small Baltic Sea region country.

## 2. Literature Review

Various approaches have been proposed to answer the question as to how and what factors influence pro-environmental and prosocial engagement. The novelty of this phenomenon complicates the identification of factors determining it. Several theories have been proposed to explain the causes and consequences of pro-environmental and prosocial engagement in sustainable consumption, such as the theory of planned behavior [30] and Stern's value–belief–norm theory [31]. According to the theory of planned behavior, an individual's decision to engage in sustainable consumption behavior is based on their behavioral intentions influenced by attitudes, norms, and perceived behavioral control [30]. Many researchers in the field of environmental psychology and behavior have used this theory to explain why consumers act sustainably [32–35].

Exploration of sustainable consumption behavior often includes environmental attitudes that indicate the consumer's thinking about environmental issues, need to change behavioral habits, and other environmental concerns [36]. However, according to Kaiser et al. [37], this theory cannot fully explain consumer engagement in sustainable consumption. To understand reasons why individuals behave sustainably, Stern [31] proposes an assessment of consumers' environmental values that lead to higher engagement in sustainable consumption [31]. Individuals with strong values tend to engage in actions that they believe can help restore those values [38]. This theory has received much attention in recent research, such as energy saving actions [39,40], green consumption behavior [31], and environmental policy support [41].

By utilizing the norm activation theory, Schwartz [42] proposes that such behavior could also be driven by personal norms, which are shaped by consumer perceptions about the consequences of behavior and their feelings of personal responsibility for those consequences [43]. This approach has been applied to a considerable number of sustainable behaviors, including recycling and household energy adaptations [44,45]. Moreover, it has been argued that perceived responsibility could also lead to unselfish behavior [46]. Nevertheless, there has been little discussion on how consumer perceived responsibility suggests a greater engagement in sustainable consumption [12]. Thus, this paper sheds new light on environmental attitudes and values, personal norms, and perceived responsibility that might influence consumers' pro-environmental and prosocial engagement in sustainable consumption and would, in turn, allow for a shift to more sustainable consumption behavior.

### 2.1. Environmental Attitudes and Engagement

Environmental attitudes are a factor which refers to the consumer's attitude towards the environment, their concern for, and understanding of, the world that surrounds them. According to the value–belief–norm theory [31], environmental attitudes are a key driver of pro-environmental behavior. Consumers with a positive environmental attitude are also more likely to have a positive view of sustainable consumption behavior, which ultimately influences their motivation to engage [47,48]. Positive environmental attitude is associated with greater engagement in sustainable consumption [49]. However, there is still little understanding of how environmental attitudes influence pro-environmental and prosocial engagement in sustainable consumption [12,18].

Environmental attitudes are assessed by analyzing a person's attitude to the reality of the limits of growth and their perceived fragility of the balance of nature, the possibility of an eco-crisis, and the inherent right of humanity to control the rest of nature [47,50]. To summarize, studies [27,49,51–54] support the idea that environmental attitudes could be positively related with the intention to engage in sustainable consumption. Based on the literature review, we suggest the following:

**Hypothesis H1.** *Environmental attitude is positively related to pro-environmental and prosocial engagement.*

## 2.2. Environmental Values and Engagement

Many sustainability scholars hold the view that values are particularly important in analyzing consumer preferences [55–57]. Previous work [58–60] has revealed the importance of environmental values (appreciation of nature and the environment) for sustainable consumption behavior. Environmental values consist of egotistic values (i.e., individualistic and materialistic consumer values) which are balanced by social altruistic values and environmentally oriented biospheric values [42]. Previous research showed that biospheric, altruistic, and egoistic values influence environmentalism and, in turn, feelings of moral obligation which result in sustainable behavior [61,62]. According to Nordlund and Garvill [63], environmental values can encourage consumers to engage in sustainable consumption. The influence of individual and group values on pro-environmental engagement has also been analyzed by Bouman, Steg, and Zawadzki [64]. Gagné [65] holds the view that group values and norms could increase prosocial engagement. Values play a key role in consumer decision making [66]. They give emotional intensity to their actions [67]. Vitell, Singhapakdi, and Thomas [68] found that consumers are guided by principles or values rather than the potential consequences of their actions in making a decision to behave sustainably.

According to the findings by Jager [69], the consumer will be motivated and involved in the process to meet their needs when they realize that the product or service is important to them and corresponds to their values. Brown and Kasser [70] found that consumers with inner motivation are more engaged in sustainable consumption than others, while Howel [61] found that inner motivation, a sense of honesty, and values were key factors in motivating environmental activists to engage in activities which are related to sustainable consumption. According to Piligrimienė et al. [12], attitudes, values, beliefs, and norms influence consumer engagement in sustainable consumption and, as a consequence, increase the buying of organic products and disposing of waste. Based on the above discussion, the following hypothesis is suggested:

**Hypothesis H2a.** *Biospheric values are positively related to pro-environmental and prosocial engagement.*

**Hypothesis H2b.** *Altruistic values are positively related to pro-environmental and prosocial engagement.*

**Hypothesis H2c.** *Egoistic values are positively related to pro-environmental and prosocial engagement.*

## 2.3. Personal Norms and Engagement

Personal norms are a kind of self-expectation, they show consumer's sense of responsibility for implementing specific actions [71], and they are considered a stronger antecedent of environmental behavior than other psychological variables (e.g., personal values, environmental concerns) [72]. Personal norms guide behavior in specific situations when individuals are aware of conditions that entail negative results for others and feel capable of averting these consequences [73]. The value–belief–norm theory [31] reveals the significance of personal norms for a person's propensity towards pro-environmental behavior. Personal norms could be conceptualized as a personal obligation related to a person's perception of responsibility for how to behave in the environment [74]. These norms are referred to as an individual's perception of fair behavior in a particular social situation [75].

Authors Stern [31] and Onel [28] argue that personal norms are one of the key determinants of pro-environmental engagement. Personal norms oblige the consumer to assume a moral obligation to behave in a certain way without harming the environment, which can help to foster a desire to engage in sustainable consumption [28]. A person with a high moral obligation is more likely to engage in pro-environmental behavior. Previous studies [76,77] have revealed that personal norms could be a significant factor influencing engagement in sustainable consumption. Therefore, we hypothesize that:

**Hypothesis H3.** *Personal norms are positively related to pro-environmental and prosocial engagement in sustainable consumption.*

### 2.4. Perceived Responsibility and Engagement

According to Luchs et al. [29] and Paço and Rodrigues [46], perceived responsibility is a factor that, as it increases, encourages consumers to engage in altruistic behavior without limiting their own actions. Perceived consumer responsibility can lead to suitable consumption choices and actions [12,29]. Perceived consumer responsibility is partly related to openness to consequences, which aims to assess the extent to which the consumer is aware of the potential environmental damage caused by their actions. This factor is associated with prosocial behavior [41], environmental activism [46], sustainable consumption choices [29,57,59], and environmentally friendly behavior [78]. The findings of research [12] revealed that consumer perceived responsibility is one of the major factors influencing engagement and shows consumer concerns not only for the present moment but also for the quality of life of future generations. Previous studies reveal that individuals with higher perceived responsibility levels increased their engagement in environmentally friendly political participation and actions [46]. Simultaneously, pro-environmental and prosocial engagement expresses an inclination to act in the interest of the environment and society. However, questions remain concerning how perceived responsibility can affect these phenomena. Thus, we suggest the following hypothesis:

**Hypothesis H4.** *Perceived responsibility is positively related to pro-environmental and prosocial engagement.*

### 2.5. Mediating Effect of Pro-Environmental and Prosocial Engagement

Few researchers have addressed the issue of consumer engagement functioning as a mediator [12,13,79] that leads to higher sustainable consumer behavior. Pro-environmental and prosocial engagement has been studied as the mediator between self-identity, values, and sustainable consumption behavior [13]. Thus, we presuppose that higher levels of pro-environmental and prosocial engagement could enable consumers to be fully aware of the potential advantages of transferring their participation and relation with environmental and social issues into sustainable consumption behavior. Pro-environmental and prosocial engagement includes three main dimensions [10,13]. The first one is the conscious attention that reveals the consumer's interest and desire to know a particular object. Another dimension of enthused participation emphasizes how invested the individual is in this process and how passionate they are about sustainable consumption behavior [80]. The social connection dimension explains how consumer interfaces with the social environment. Pro-environmental and prosocial engagement is a specific psychological state that enables consumers to better understand and evaluate preferences during the decision making process and helps to maintain focus on behavior change. We propose that pro-environmental and prosocial engagement functions as a mechanism through which chosen factors have a bigger influence on sustainable consumption behavior. Consequently, we postulate the following hypotheses:

**Hypothesis H5.** *Pro-environmental and prosocial engagement mediates the link between environmental attitudes and sustainable consumption behavior, thus, reinforcing the positive effects on sustainable consumption behavior.*

**Hypothesis H6a.** *Pro-environmental and prosocial engagement mediates the link between biospheric values and sustainable consumption behavior, thus, reinforcing the positive effects on sustainable consumption behavior.*

**Hypothesis H6b.** *Pro-environmental and prosocial engagement mediates the link between altruistic values and sustainable consumption behavior, thus, reinforcing the positive effects on sustainable consumption behavior.*

**Hypothesis H6c.** *Pro-environmental and prosocial engagement mediates the link between egoistic values and sustainable consumption behavior, thus, reinforcing the positive effects on sustainable consumption behavior.*

**Hypothesis H7.** *Pro-environmental and prosocial engagement mediates the link between personal norms and sustainable consumption behavior, thus, reinforcing the positive effects on sustainable consumption behavior.*

**Hypothesis H8.** *Pro-environmental and prosocial engagement mediates the link between perceived responsibility and sustainable consumption behavior, thus, reinforcing the positive effects on sustainable consumption behavior.*

*2.6. Engagement and Sustainable Consumption Behavior*

Consumer engagement in an environmental and social context can also be seen as a form of behavior [81], as concern for emerging problems [82], or as an interest of community members to support others and participate voluntarily in joint activities [83]. However, according to researchers [10], pro-environmental and prosocial engagement can link both processes, psychological and participatory. According Čapienė et al. [18], pro-environmental and prosocial engagement can be described as a consumer's psychological state that expresses a desire to act in the interest of the environment and the public. According to De Groot and Steg [84], pro-environmental engagement can also be treated as a case of prosocial engagement, as in both cases the consumers involved do not benefit personally, and the greatest efforts are dedicated for the benefit of others. These forms of engagement intertwine, since efforts to preserve the environment also have positive consequences for others. In addition, it has been established that pro-environmental behavior is strongly associated with prosocial behavior because individuals direct their actions to promote the health and well-being of other individuals now and in the future [85]. The engagement itself is a phenomenon described by a wide range of aspects. In this study, we consider pro-environmental and prosocial engagement to include a conscious focus on environmental activities that benefit society.

Nevertheless, pro-environmental and prosocial engagement does not necessarily have to be focused on sustainable consumption. An individual exhibiting a high level of pro-environmental or prosocial engagement may also engage in other activities that are related to the environment and are of benefit to society, such as river cleaning, tree planting, and so on. Within the scope of this study, we aim to explain how this engagement could shift to long lasting sustainable consumption behavior, as the results of the SOR model show. In other words, it is the real outcome of the whole sustainable consumption decision making process that overcomes the challenge of the intention–action gap. Based on the above discussion, we propose the following hypothesis:

**Hypothesis H9.** *Pro-environmental and prosocial consumer engagement in sustainable consumption is positively related to sustainable consumption behavior.*

Building on these hypotheses, we propose a theoretical model presented in Figure 1, depicting the links between environmental attitudes and values, personal norms, perceived responsibility, pro-environmental and prosocial consumer engagement in sustainable consumption, and sustainable consumption behavior.

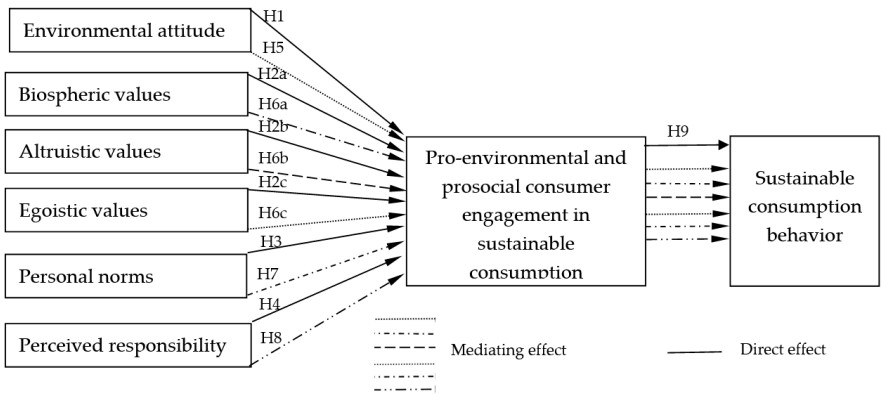

**Figure 1.** Research model.

## 3. Methodology

### 3.1. Sample

The current study aims to explore the relationships between environmental attitudes and values, personal norms, perceived responsibility, pro-environmental and prosocial engagement in sustainable consumption, and sustainable consumption behavior. Using the survey method, a questionnaire was developed, pretested, and distributed among individuals over 18 years of age in Lithuania. The use of this method is pertinent in testing the relationships between environmental attitudes and values, personal norms, perceived responsibility, pro-environmental and prosocial engagement in sustainable consumption, and sustainable consumption behavior [86]. The non-statistical comparative method was used to determine the sample size. To do so, we found 12 comparable studies that investigate the relationships of at least 2 constructs of the current study. Based on that, the sample size estimate is 390 respondents. Nevertheless, in this study, we aimed to reduce the error to ±3.5%. According to Cohen et al. [87], the sample size of the study in this case should be at least 800 respondents.

The study opted for a non-random quota sampling method since it is not possible to obtain a sampling frame or a population list. The study is based on demographic criteria but is not randomly selected (see Table 1). Mutually unrelated gender and age quotas were applied to this survey, which were calculated according to the population composition statistics provided on the official statistics portal of Statistics of Lithuania (osp.stat.gov.lt). The study sample maintained a gender ratio relative to the proportion in the population, but there are more younger representatives in the sample.

**Table 1.** Quota population distribution by gender and age.

|  | Population | Population Distribution, % | Research Respondent Distribution, % | Number of Fully Completed Questionnaires |
|---|---|---|---|---|
| Men | 868,288 | 47% | 47.7% | 431 |
| Women | 903,729 | 53% | 52.3% | 473 |
| 16–29 | 389,843 | 18.6% | 25.3% | 229 |
| 30–39 | 372,123 | 15.9% | 22.6% | 204 |
| 40–49 | 354,406 | 15.6% | 20% | 181 |
| 50–59 | 301,242 | 17.6% | 13.1% | 118 |
| 60 and over | 354,403 | 32.4% | 19% | 172 |
| Total |  |  |  | **904** |

A total of 1165 questionnaires were completed during the survey, but 261 questionaries were removed due to incorrect and malicious completion. Therefore, the final sample of the study consists of 904 respondents. This exceeds the expected number of respondents and allows us to reduce the margin of error to ±3.25%.

*3.2. Data Collection*

The data was collected from 1 October 2020 to 30 November 2020, during the pandemic, with the help of a digital platform. The link to the survey has been disseminated through various channels, such as social networks (Facebook and Instagram), and via applications to companies, organizations, third-age universities, and educational institutions.

The final sample was composed of 47.7% men and 52.3% women. The age range was from 18 to 92 years, with a mean age of 47 years. The breakdown of respondents in terms of age was age 18–29 ($N = 229$), age 30–39 ($N = 294$), age 40–49 ($N = 181$), age 50–59 ($N = 118$), and age >60 ($N = 172$). Most of respondents represented underwent higher education (51.5%). More than half (58.6%) of respondents indicated that they live like most Lithuanian people.

*3.3. Measures*

To measure constructs, we used previously established and validated scales (Appendix A). As in the case of Dunlap et al. [47], a 7-point Likert-type scale (1 being "strongly disagree" and 7 being "strongly agree") was adopted to measure environmental attitudes (15 items). For example, we asked respondents to evaluate their agreement or disagreement with such statements as "we are approaching the limit of the number of people the Earth can support" or "humans have the right to modify the natural environment to suit their needs.". To measure environmental values, we used 13 items adopted from Howell [61] and Van Riper and Kylie [62], as based on Schwartz [65]. From these, four items reflect biospheric values, four reflect altruistic values, and five reflect egoistic values. Respondents were asked to rate the importance of each value on a 7-point scale from 1 (not important at all) to 7 (very important). Some examples of the items are "social power: control over others, dominance" and "unity with nature: fitting into nature". Personal norms were measured using Vining and Ebreo's [45] proposed five items. Respondents expressed their opinion using a 7-point Likert scale (1 being "strongly disagree" and 7 being "strongly agree") on such items as "I feel obliged to sort a large proportion of household waste" or "I would sort out household waste, whether or not I get paid for it". The perceived responsibility construct was measured using a 7-point Likert scale that contained seven items proposed by Paço and Rodrigues [46]. Some examples are "environmental protection starts with me" and "environmental protection is the responsibility of my government, not me". Eight items capturing pro-environmental and prosocial engagement were borrowed from Kadic-Maglajlic et al. [13]. Respondents were asked to measure their agreement or disagreement using a 7-point Likert scale (one item example was "I like to learn more about [environmentally/socially]-friendly behavior"). Finally, for the sustainable consumption behavior construct, we used 23 items proposed by Quoquab, Mohammad, and Sukari [7].

## 4. Results

*4.1. Exploratory Factor Analysis*

We conducted an exploratory factor analysis to evaluate the suitability of scales and to identify the structure of constructs. Each scale was verified separately using the principal components method of extraction (varimax rotation). The results of EFA and respective Cronbach alphas are presented in Table 2.

**Table 2.** Exploratory factor analysis.

| Factor | No of Items | KMO | Range of Factor Loading | Variance Explained by Each Factor, % | Cronbach Alpha |
|---|---|---|---|---|---|
| Environmental attitude | 11 | 0.819 | 0.852–0.737 | 19.21 | 0.730 |
| Biospheric values | 4 | 0.823 | 0.676–0.864 | 12.01 | 0.890 |
| Altruistic values | 3 | 0.793 | 0.651–0.839 | 6.79 | 0.821 |
| Egoistic values | 5 | 0.693 | 0.671–0.827 | 24.91 | 0.794 |

**Table 2.** *Cont.*

| Factor | No of Items | KMO | Range of Factor Loading | Variance Explained by Each Factor, % | Cronbach Alpha |
|---|---|---|---|---|---|
| Personal norms | 5 | 0.817 | 0.600–0.851 | 25.71 | 0.809 |
| Perceived responsibility | 5 | 0.829 | 0.721–0.854 | 26.33 | 0.842 |
| Pro-environmental and prosocial engagement | 8 | 0.866 | 0.600–0.856 | 63.63 | 0.893 |
| Sustainable consumption behaviour | 18 | 0.813 | 0.608–0.747 | 27.93 | 0.863 |

After performing the analysis of the structure of conceptual model constructs, we identified factors that would allow for the modification of original variables. It should be noted that the construct of pro-environmental and prosocial engagement has been modified as the factor analysis showed one factor; thus, in further analysis, this construct is analyzed as a one-dimensional construct. Instead of the theoretically predicted three dimensions, the sustainable consumption behavior scale has obtained a one-dimensional scale. The value of KMO is 0.813, thus, the analysis of the variables is well suited. Environmental attitude, personal norms, and perceived responsibility constructs are analyzed as a reliable one-dimensional scale.

*4.2. Correlation Analysis*

Correlation analysis was performed to check the existence of statistically significant relationships between analyzed constructs (Table 3).

**Table 3.** Means, standard deviations, and correlations among variables.

| Variables | 1. | 2. | 3. | 4. | 5. | 6. | 7. |
|---|---|---|---|---|---|---|---|
| 1. Environmental attitude | 1 | | | | | | |
| 2. Biospheric values | 0.362 ** | 1 | | | | | |
| 3. Altruistic values | 0.283 ** | 0.498 ** | 1 | | | | |
| 4. Egoistic values | 0.04 | 0.132 ** | 0.116 ** | 1 | | | |
| 5. Personal norms | 0.297 ** | 0.554 ** | 0.422 ** | −0.044 | 1 | | |
| 6. Perceived responsibility | 0.297 ** | 0.539 ** | 0.327 ** | 0.093 ** | 0.536 ** | | |
| 7. Pro-environmental and prosocial engagement | 0.325 ** | 0.530 ** | 0.297 ** | 0.131 ** | 0.521 ** | 0.586 ** | 1 |
| 8. Sustainable consumption behavior | 0.314 ** | 0.487 ** | 0.346 ** | 0.136 ** | 0.482 ** | 0.510 ** | 0.694 ** |

** $p < 0.01$.

A further correlation analysis (Table 3) revealed that almost all variables are statistically significantly related to each other ($p < 0.001$ or $p < 0.05$). All statistically significant relationships are positive with some differences in extent; pro-environmental and prosocial engagement most strongly correlates with the scale of sustainable consumption. The strongest correlation is between pro-environmental and prosocial engagement and perceived responsibility, biospheric values, and personal norms—their Spearman's rank correlation coefficient is $p > 0.05$. Additionally, all factors are statistically significantly related to the scale of sustainable consumption behavior.

A Mann–Whitney U test and correlation analysis were performed to check if there are statistically significant differences in the scores to check for the existence of statistically significant relationships between the analyzed constructs and respondents' age, gender, education, and subjective financial situation. A statistically significant relationship between respondents' age and the following variables was identified as follows: sustainable consumption behavior ($r = 0.19$, $p < 0.05$), pro-environmental and prosocial engagement ($r = 0.125$, $p < 0.05$). Moreover, a weaker relationship was identified between age and altruistic values ($r = −0.088$, $p < 0.05$). This analysis found a reverse statistically significant relationship between environmental attitude and age ($r = −0.194$, $p < 0.05$). A statistically

significant relationship was not found between age and the following constructs: perceived responsibility, biospheric values, egoistic values, and personal norms. In addition, a statistically significant relationship was identified between the respondents' gender (i.e., woman) and such variables as personal norms, perceived responsibility, biospheric values, altruistic values, environmental attitudes, pro-environmental and prosocial engagement, and sustainable consumption behavior (Table 4). Furthermore, the strongest statistically significant relationship was found between the respondents' education level and personal norms (r = 0.173, $p < 0.05$). However, the subjective financial situation variable showed no correlation with variables, with the exception of perceived responsibility.

**Table 4.** The $p$ values and Spearman's rank correlation coefficients for non-parametric tests of independent variables of control variables and scales.

| Scale | Age Group | | Gender (Female) | | Education | | Subjective Financial Situation | |
|---|---|---|---|---|---|---|---|---|
| | Kruskal–Wallis Test ($p$) | Spearman's Rank Correlation Coefficient | Mann–Whitney U Test ($p$) | Spearman's Rank Correlation Coefficient | Kruskal–Wallis Test ($p$) | Spearman's Rank Correlation Coefficient | Kruskal–Wallis Test ($p$) | Spearman's Rank Correlation Coefficient |
| Environmental attitude | 0.000 | −0.194 * | 0.000 | 0.161 * | 0.669 | −0.043 | 0.376 | −0.006 |
| Biospheric values | 0.001 | 0.002 | 0.000 | 0.196 * | 0.084 | 0.093 * | 0.591 | 0.015 |
| Altruistic values | 0.024 | −0.088 * | 0.000 | 0.195 * | 0.068 | 0.081 * | 0.783 | 0.014 |
| Egoistic values | 0.025 | 0.061 | 0.004 | −0.095 * | 0.001 | −0.110 * | 0.476 | 0.007 |
| Personal norms | 0.002 | −0.051 | 0.000 | 0.310 * | 0.000 | 0.173 * | 0.006 | 0.048 |
| Perceived responsibility | 0.010 | −0.032 | 0.000 | 0.210 * | 0.033 | 0.100 * | 0.002 | 0.072 * |
| Pro-environmental and prosocial engagement | 0.001 | 0.125 * | 0.000 | 0.188 * | 0.066 | 0.082 * | 0.006 | 0.043 |
| Sustainable consumption behaviour | 0.000 | 0.190 * | 0.000 | 0.188 * | 0.007 | 0.054 | 0.002 | −0.001 |

* $p < 0.05$.

Based on the results of the correlation analysis, it can be concluded that hypotheses H1, H2a, H2b, H2c, H3, and H4, stating that environmental attitude, biospheric values, altruistic values, egoistic values, personal norms, and perceived responsibility are positively related to pro-environmental and prosocial engagement, were confirmed. The findings also show that hypothesis H9 (pro-environmental and prosocial consumer engagement in sustainable consumption is positively related to sustainable consumption behavior) was confirmed.

*4.3. Mediation Analysis*

Aiming to identify indirect effects further, we performed a regression-based mediation analysis using an SPSS PROCESS model to test hypotheses H5, H6, H7, and H8, as presented in Figure 2.

The environmental attitudes mediation model shows that all three parametric regression models are statistically significant ($p < 0.05$ for all F tests). The overall influence of environmental attitude on sustainable consumption behavior is statistically significant (c = 0.38, $p < 0.05$). It was found that the direct influence of environmental attitude on sustainable consumption behavior (c′) is statistically significant (c′ = 0.14, $p < 0.05$). Moreover, the indirect–mediation–pathway a × b is statistically significant (a × b = 0.24, $p < 0.05$). It can be stated that more than half of the total influence of environmental attitudes (c = 0.38, $p < 0.05$) on sustainable consumption behavior is mediated through pro-environmental and prosocial involvement (a × b = 0.24, $p < 0.05$). Thus, hypothesis H5 stands. The results of the indirect effect of environmental attitude on sustainable consumption behavior through pro-environmental and prosocial engagement are summarized in Table 5.

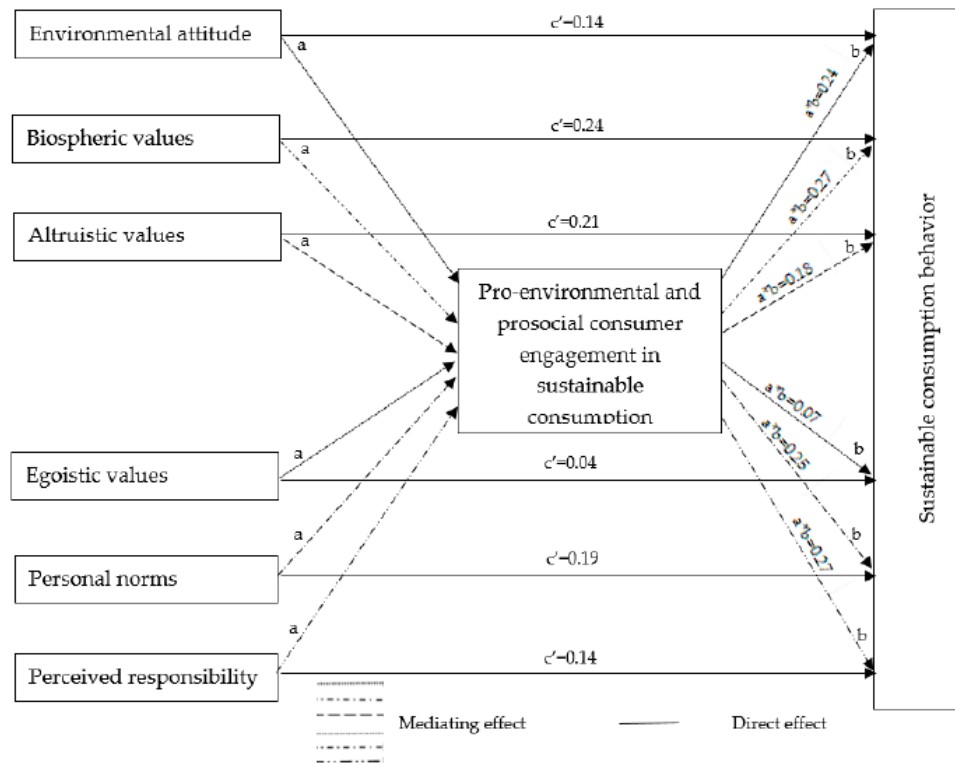

**Figure 2.** Mediation analysis results.

**Table 5.** Mediation analysis, as follows: indirect effect of environmental attitudes on sustainable consumption behavior through pro-environmental and prosocial engagement.

| Regressor | M (a) Pro-Environmental and Prosocial Engagement | | | | Y (c′, b) Sustainable Consumption Behavior | | | | Y (c) Sustainable Consumption Behavior | | | | Y (a × b) Sustainable Consumption Behavior | | | |
|---|---|---|---|---|---|---|---|---|---|---|---|---|---|---|---|---|
| | Path | Coeff. | SE | *p* | Path | Coeff. | SE | *p* | Path | Coeff. | SE | *p* | Path | Coeff. | SE | PI95% |
| **X:EA** | a | 0.46 | 0.05 | 0.00 | c′ | 0.14 | 0.04 | 0.00 | c | 0.383 | 0.042 | 0.00 | a × b | 0.24 * | 0.028 | [0.187; 0.295] |
| **M:PPE** | - | - | - | - | b | 0.52 | 0.02 | 0.00 | | | | | - | - | - | - |
| **Intercept** | $i_M$ | 2.53 | 0.27 | 0.00 | $i_Y$ | 2.05 | 0.19 | 0.00 | $i_Y$ | 3.366 | 0.222 | 0.00 | | - | - | - |
| **Model summary** | $R^2 = 0.11$; $F(1.902) = 83.762$, $p = 0.000$ | | | | $R^2 = 0.51$; $F(2.901) = 348.016$, $p = 0.000$ | | | | $R^2 = 0.12$; $F(1.902) = 83.696$, $p = 0.000$ | | | | - | | | |

\* Here, $p < 0.05$; Abbreviations are as follows: EA—environmental attitude; PPE—pro-environmental and prosocial engagement; SCB—sustainable consumption behavior.

The results of the indirect effect of biospheric values on sustainable consumption behavior through pro-environmental and prosocial engagement (hypothesis H6a) are summarized in Table 6. The biospheric values mediation model shows that all three parametric regression models are statistically significant ($p < 0.05$ for all F tests). The overall influence of biosphere values on sustainable consumption behavior is statistically significant (c = 0.52, $p < 0.05$). The direct influence of biospheric values on sustainable consumption behavior (c′) is statistically significant (c′ = 0.24, $p < 0.05$). The indirect–mediation–pathway a × b is statistically significant (a × b = 0.27, $p < 0.05$). It can be stated that slightly more than half of the total influence of the independent variable biospheric value (c = 0.52, $p < 0.05$) on sustainable consumption behavior is mediated through pro-environmental and prosocial engagement (a × b = 0.27, $p < 0.05$). Consequently, hypothesis H6a stands.

The results of the indirect effect of biospheric values on sustainable consumption behavior through pro-environmental and prosocial engagement (hypothesis H6a) are summarized in Table 6. The biospheric values mediation model shows that all three

parametric regression models are statistically significant ($p < 0.05$ for all F tests). The overall influence of biosphere values on sustainable consumption behavior is statistically significant (c = 0.52, $p < 0.05$). The direct influence of biosphere values on sustainable consumption behavior (c′) is statistically significant (c′ = 0.24, $p < 0.05$). The indirect–mediation–pathway a × b is statistically significant (a × b = 0.27, $p < 0.05$). It can be stated that slightly more than half of the total influence of the independent variable biospheric values (c = 0.52, $p < 0.05$) on sustainable consumption behavior is mediated through pro-environmental and prosocial engagement (a × b = 0.27, $p < 0.05$). Consequently, hypothesis H6a stands.

**Table 6.** Mediation analysis, as follows: indirect effect of biospheric values on sustainable consumption behavior through pro-environmental and prosocial engagement.

| Regressor | M (a) Pro-Environmental and Prosocial Engagement | | | | Y (c′, b) Sustainable Consumption Behavior | | | | Y (c) Sustainable Consumption Behavior | | | | Y (a × b) Sustainable Consumption Behavior | | | |
|---|---|---|---|---|---|---|---|---|---|---|---|---|---|---|---|---|
| | Path | Coeff. | SE | p | Path | Coeff. | SE | p | Path | Coeff. | SE | p | Path | Coeff. | SE | PI95% |
| **X:BV** | a | 0.62 | 0.04 | 0.00 | c′ | 0.24 | 0.03 | 0.00 | c | 0.515 | 0.027 | 0.00 | a × b | 0.27 * | 0.022 | [0.232; 0.319] |
| **M:PPE** | - | - | - | b | 0.44 | 0.02 | 0.00 | - | - | - | - | - | - | - | - | |
| **Intercept** | $i_M$ | 1.16 | 0.22 | 0.00 | $i_Y$ | 1.71 | 0.16 | 0.00 | $i_Y$ | 2.225 | 0.166 | 0.00 | | - | - | - |
| **Model summary** | $R^2$ = 0.29; F(1.902) = 297.676, $p$ = 0.000 | | | | $R^2$ = 0.54; F(2.901) = 431.040, $p$ = 0.000 | | | | $R^2$ = 0.32; F(1.902) = 365.619, $p$ = 0.000 | | | | - | | | |

\* Here, $p < 0.05$; Abbreviations are as follows: BV—biospheric values; PPE—pro-environmental and prosocial engagement; SCB—sustainable consumption behavior.

The results of the indirect effect of altruistic values on sustainable consumption behavior through pro-environmental and prosocial engagement (hypothesis H6b) are summarized in Table 7. Altruistic values explain a small proportion of the median pro-environmental and prosocial engagement in the mean distribution ($R^2$ = 0.1, path a), as well as a small mean variance in the dependent variable sustainable consumption behavior (R2 = 0.18, path c). The overall influence of altruistic values on sustainable consumption behavior is statistically significant (c = 0.39, $p < 0.05$). The direct influence of biosphere values on sustainable consumption behavior (c′) is statistically significant (c′ = 0.21, $p < 0.05$). The indirect–mediation–pathway a × b is statistically significant (a × b = 0.18, $p < 0.05$). It can be stated that slightly less than half of the total influence of the independent variable altruistic values (c = 0.39, $p < 0.05$) on sustainable consumption behavior is mediated through pro-environmental and prosocial engagement (a × b = 0.18, $p < 0.05$). Thus, hypothesis H6b stands.

**Table 7.** Mediation analysis, as follows: indirect effect of altruistic values on sustainable consumption behavior through pro-environmental and prosocial engagement.

| Regressor | M (a) Pro-Environmental and Prosocial Engagement | | | | Y (c′, b) Sustainable Consumption Behavior | | | | Y (c) Sustainable Consumption Behavior | | | | Y (a × b) Sustainable Consumption Behavior | | | |
|---|---|---|---|---|---|---|---|---|---|---|---|---|---|---|---|---|
| | Path | Coeff. | SE | p | Path | Coeff. | SE | p | Path | Coeff. | SE | p | Path | Coeff. | SE | PI95% |
| **X:AV** | a | 0.35 | 0.04 | 0.00 | c′ | 0.21 | 0.03 | 0.00 | c | 0.385 | 0.037 | 0.00 | a × b | 0.18 * | 0.021 | [0.135; 0.220] |
| **M:PPE** | - | - | - | b | 0.50 | 0.02 | 0.00 | - | - | - | - | - | - | - | - | |
| **Intercept** | $i_M$ | 2.72 | 0.27 | 0.00 | $i_Y$ | 1.59 | 0.17 | 0.00 | $i_Y$ | 2.953 | 0.233 | 0.00 | | - | - | - |
| **Model summary** | $R^2$ = 0.10; F(1.901) = 67.277, $p$ = 0.000 | | | | $R^2$ = 0.54; F(2.900) = 429.772, $p$ = 0.000 | | | | $R^2$ = 0.18; F(1.901) = 109.376, $p$ = 0.000 | | | | - | | | |

\* Here, $p < 0.05$; Abbreviations are as follows: AV—altruistic values; PPE—pro-environmental and prosocial engagement; SCB—sustainable consumption behavior.

The results of the indirect effect of altruistic values on sustainable consumption behavior through pro-environmental and prosocial engagement (hypothesis H6b) are summarized in Table 7. Altruistic values explain the small proportion of the median pro-environmental and prosocial engagement in the mean distribution (R2 = 0.1, path a), as well as the small mean variance in the dependent variable sustainable consumption behavior

($R^2$ = 0.18, path c). The overall influence of altruistic values on sustainable consumption behavior is statistically significant (c = 0.39, $p < 0.05$). The direct influence of biosphere values on sustainable consumption behavior (c′) is statistically significant (c′ = 0.21, $p < 0.05$). The indirect–mediation–pathway a × b is statistically significant (a × b = 0.18, $p < 0.05$). It can be stated that slightly less than half of the total influence of the independent variable altruistic values (c = 0.39, $p < 0.05$) on sustainable consumption behavior is mediated through pro-environmental and prosocial engagement (a × b = 0.18, $p < 0.05$). Thus, hypothesis H6b stands.

The results of the indirect effect of egoistic values on sustainable consumption behavior through pro-environmental and prosocial engagement (hypothesis H6c) are summarized in Table 8. The direct influence of egoistic values on sustainable consumption behavior is statistically significant (c′ = 0.04, $p = 0.05$), as is the mediator's pro-environmental and prosocial engagement (b = 0.55, $p < 0.05$). The indirect–mediation–pathway a × b is statistically significant (a × b = 0.07, $p < 0.05$). It can be stated that more than half of the total influence of the independent variable egoistic value (c = 0.11, $p < 0.05$) on sustainable consumption behavior is mediated through pro-environmental and prosocial engagement (a × b = 0.07, $p < 0.05$). Thus, hypothesis H6c was confirmed.

**Table 8.** Mediation analysis, as follows: indirect effect of egoistic values on sustainable consumption behavior through pro-environmental and prosocial engagement.

| Regressor | M (a) Pro-Environmental and Prosocial Engagement | | | | Y (c′, b) Sustainable Consumption Behavior | | | | Y (c) Sustainable Consumption Behavior | | | | Y (a × b) Sustainable Consumption Behavior | | | |
|---|---|---|---|---|---|---|---|---|---|---|---|---|---|---|---|---|
| | Path | Coeff. | SE | p | Path | Coeff. | SE | p | Path | Coeff. | SE | p | Path | Coeff. | SE | PI95% |
| **X:EV** | a | 0.12 | 0.04 | 0.00 | c′ | 0.04 | 0.02 | 0.05 | c | 0.109 | 0.033 | 0.00 | a × b | 0.07 * | 0.022 | [0.024; 0.109] |
| **M:PPE** | - | - | - | - | b | 0.55 | 0.02 | 0.00 | - | - | - | - | - | - | - | - |
| **Intercept** | $i_M$ | 4.35 | 0.19 | 0.00 | $i_Y$ | 2.42 | 0.15 | 0.00 | $i_Y$ | 4.819 | 0.167 | 0.00 | | - | - | - |
| **Model summary** | $R^2$ = 0.01; F(1.901) = 9.214, $p = 0.003$ | | | | $R^2$ = 0.50; F(2.900) = 335.369, $p = 0.000$ | | | | $R^2$ = 0.02; F(1.901) = 11.168, $p = 0.001$ | | | | | - | | |

* Here, $p < 0.05$; Abbreviations are as follows: EV—egoistic values; PPE—pro-environmental and prosocial engagement; SCB—sustainable consumption behavior.

The results of the indirect effect of personal norms on sustainable consumption behavior through pro-environmental and prosocial engagement (hypothesis H7) are summarized in Table 9. The direct influence of personal norms on sustainable consumption behavior (c′) is statistically significant (c′ = 0.19, $p < 0.05$). The indirect–mediation–pathway a × b is statistically significant (a × b = 0.25, $p < 0.05$). It can be stated that more than half of the total influence of the independent variable personal norm (c = 0.44, $p < 0.05$) on sustainable consumption behavior is mediated through pro-environmental and prosocial engagement (a × b = 0.25, $p < 0.05$). Thus, hypothesis H7 stands.

**Table 9.** Mediation analysis, as follows: indirect effect of personal norms on sustainable consumption behavior through pro-environmental and prosocial engagement.

| Regressor | M (a) Pro-Environmental and Prosocial Engagement | | | | Y (c′, b) Sustainable Consumption Behaviour | | | | Y (c) Sustainable Consumption Behaviour | | | | Y (a×b) Sustainable Consumption Behaviour | | | |
|---|---|---|---|---|---|---|---|---|---|---|---|---|---|---|---|---|
| | Path | Coeff. | SE | p | Path | Coeff. | SE | p | Path | Coeff. | SE | p | Path | Coeff. | SE | PI95% |
| **X:PN** | a | 0.55 | 0.03 | 0.00 | c′ | 0.19 | 0.03 | 0.00 | c | 0.439 | 0.025 | 0.00 | a × b | 0.25 * | 0.018 | [0.212; 0.284] |
| **M:PPE** | - | - | - | - | b | 0.45 | 0.02 | 0.00 | - | - | - | - | - | - | - | - |
| **Intercept** | $i_M$ | 1.79 | 0.16 | 0.00 | $i_Y$ | 2.07 | 0.13 | 0.00 | $i_Y$ | 2.864 | 0.146 | 0.00 | | - | - | - |
| **Model summary** | $R^2$ = 0.32; F(1.902) = 394.969, $p = 0.000$ | | | | $R^2$ = 0.54; F(2.901) = 407.664, $p = 0.000$ | | | | $R^2$ = 0.32; F(1.902) = 312.946, $p = 0.000$ | | | | | - | | |

* Here, $p < 0.05$; Abbreviations are as follows: PN—personal norms; PPE—pro-environmental and prosocial engagement; SCB—sustainable consumption behavior.

The results of the indirect effect of perceived responsibility on sustainable consumption behavior through pro-environmental and prosocial engagement (hypothesis H8) are summarized in Table 10. The direct influence of perceived responsibility on sustainable consumption behavior is statistically significant ($c' = 0.14$, $p < 0.05$). The indirect–mediation–pathway $a \times b$ is statistically significant ($a \times b = 0.27$, $p < 0.05$). It can be stated that more than half of the total influence of the independent variable perceived responsibility ($c = 0.41$, $p < 0.05$) on sustainable consumption behavior is mediated through pro-environmental and prosocial engagement ($a \times b = 0.27$, $p < 0.05$). Consequently, hypothesis H8 was confirmed.

**Table 10.** Mediation analysis, as follows: indirect effect of perceived responsibility on sustainable consumption behavior through pro-environmental and prosocial engagement.

| Regressor | M (a) Pro-Environmental and Prosocial Engagement | | | | Y (c′, b) Sustainable Consumption Behavior | | | | Y (c) Sustainable Consumption Behavior | | | | Y (a × b) Sustainable Consumption Behavior | | | |
|---|---|---|---|---|---|---|---|---|---|---|---|---|---|---|---|---|
| | Path | Coeff. | SE | p | Path | Coeff. | SE | p | Path | Coeff. | SE | p | Path | Coeff. | SE | PI95% |
| **X:PR** | a | 0.58 | 0.03 | 0.00 | c′ | 0.14 | 0.03 | 0.00 | c | 0.412 | 0.026 | 0.00 | a × b | 0.27 * | 0.019 | [0.236; 0.312] |
| **M:PPE** | - | - | - | b | 0.47 | 0.03 | 0.00 | | - | - | - | | - | - | - |
| **Intercept** | $i_M$ | 1.93 | 0.15 | 0.00 | $i_Y$ | 2.32 | 0.13 | 0.00 | $i_Y$ | 3.224 | 0.138 | 0.00 | | - | - | - |
| **Model summary** | $R^2 = 0.36$; F(1.902) = 409.396, $p = 0.000$ | | | | $R^2 = 0.51$; F(2.901) = 365.878, $p = 0.000$ | | | | $R^2 = 0.29$; F(1.902) = 260.995, $p = 0.000$ | | | | - | | | |

\* Here, $p < 0.05$; Abbreviations are as follows: PR—Perceived responsibility; PPE—pro-environmental and prosocial engagement; SCB—sustainable consumption behavior.

## 5. Discussion

This study examines links between environmental attitudes and values, personal norms, perceived responsibility, pro-environmental and prosocial engagement in sustainable consumption, and sustainable consumption behavior. A conceptual model has been created based on the logic of the SOR model, where environmental attitudes and values, personal norms, and perceived responsibility are the stimuli, pro-environmental and prosocial engagement in sustainable consumption is the mechanism, and sustainable consumption behavior is the response. We have focused on pro-environmental and prosocial aspects of engagement. Pro-environmental engagement is strongly related to prosocial engagement, as consumers do not benefit from it individually and most of their actions are directed at benefitting others. The literature review reveals that this approach is not widely investigated but is very effective in disclosing these links and the role of engagement as the mediator. This result ties in well with a previous study [13]. It is important to note that the present research relies on the idea that higher levels of pro-environmental and prosocial engagement could enable consumers to orient themselves towards sustainable consumption behavior.

As expected, our study found that environmental attitudes, environmental values (biospheric, altruistic, and egoistic), personal norms, and perceived responsibility are positively related with pro-environmental and prosocial engagement in sustainable consumption. This is in line with the results of previous research [12,27,29,46,76]. In addition, our findings demonstrate that personal norms and perceived responsibility have the most significant relationship with pro-environmental and prosocial engagement within our sample. This is in line with the results of previous studies [13,29,88–92]. In general, these results imply that consumers who have high levels of self-expectation and a sense of responsibility for implementing specific actions and openness to consequences are more engaged in sustainable consumption.

While testing the indirect effect of stimuli (i.e., perceived responsibility, biospheric values, egoistic values, personal norms, and environmental attitudes) on sustainable consumption behavior via engagement, it has been identified that independent variables (perceived responsibility, biospheric values, egoistic values, personal norms, and environmental attitude) have statistically significant positive effects on sustainable consumption behavior. These results support the statement that pro-environmental and prosocial en-

gagement acts as a mediator and fosters consumers' sustainable consumption behavior. That is consistent with previous research [13,27,91]. However, slightly less than half of the total influence of the altruistic values on sustainable consumption behavior is mediated by pro-environmental and prosocial engagement. These findings suggest that altruistic values are less mediated with engagement than biospheric or egoistic values. This substantiates previous findings in the literature that indicate the fact that altruistic values have stronger direct influence on sustainable consumption behavior [93].

Our data also support a significant and strong relation between consumers' pro-environmental and prosocial engagement and sustainable consumption behavior. This substantiates previous findings which analyzed consumer engagement as a factor in promoting sustainable consumption behavior [14–17].

Most studies analyzing pro-environmental and prosocial engagement have been conducted in groups of young people [13,94]. In our study, we included consumers from various age groups. The current research revealed that older (over 50-years-old) consumers are more engaged, and consume more sustainably, but more positive environmental attitudes were found in the younger consumer group (16–39 years). This could be explained by the fact that older respondents are more attached to their place of residence, may even identify with it, and have better environmental knowledge or good intentions with regards to sustainable consumption behavior. Furthermore, the younger respondents have more positive environmental attitudes, but that does not lead to higher engagement and sustainable behavior in Lithuania. From this standpoint, further research questions should focus on intention and real action, and how sustainably different age groups would act in real situations (via the use of an experiment). In addition, the present study confirmed that women are more concerned about pro-environmental and prosocial issues. This corresponds well with the study of Costa Pinto et al. [95]. Our research confirms previous research [96] that shows consumers with higher education to distinguish themselves as individuals those perceive their fair behavior in a particular social situation. Thus, having stronger personal norms, educated consumers are more engaged and behave, therefore, in more sustainable ways.

## 6. Conclusions

The current study expands the literature on sustainable consumption behavior while explaining links between under-researched internal factors and pro-environmental and prosocial engagement in the context of the SOR model. Previous studies have focused mostly on antecedents of sustainable consumer behavior but have not delved into the mediation effect of pro-environmental and prosocial engagement as the phenomenon that could decrease the gap between attitude and actual behavior. Our study allows for the confirmation that environmental attitudes, environmental values, personal norms, and perceived responsibility could be reinforced within engagement and, in turn, foster sustainable behavior among Lithuanian consumers.

Future research should expand the current understanding of the mediating effect of pro-environmental and prosocial engagement, as well as its stimuli and the result, namely sustainable consumer behavior. The nature and dominance of the stimuli of consumers' pro-environmental and prosocial engagement in sustainable consumption may vary across countries and cultures. It should be noted that this empirical study was conducted in only one country, Lithuania. There is a probability that cultural social disparities may reflect other factors of consumers' pro-environmental and prosocial engagement.

Furthermore, the research was performed using the method of non-random quota sampling and and cross-sectional research design, which leads to the inaccuracy of certain data. Thus, we cannot present generalizable conclusions about the population.

The study was conducted during the pandemic. People at this period felt more sensitive to environmental and social problems, which could have led to the confirmation of highlighted hypotheses.

The study was also conducted without detailing the context. It would be useful to carry out research on specific areas of sustainable consumption, such as food, housing, mobility, clothing, etc. In addition, the causal research design would add value for further research.

The research findings have some managerial social implications. For example, policy-makers, community representatives, or leaders could adapt the knowledge of factors which have the greatest impact on pro-environmental and prosocial engagement, which in turn promotes sustainable consumption behavior, by organizing social promotion campaigns or providing guidance on project objectives. Representatives of socially responsible businesses could apply this knowledge by creating product packaging, narratives, and advertising, providing information at the point of purchase or on the company's website, or planning social marketing communications and public relations campaigns.

**Author Contributions:** Conceptualization, A.Č., A.R. and K.K.; methodology, A.R.; software, A.Č., A.R. and K.K.; validation, A.Č. and A.R.; formal analysis, A.Č. and A.R.; investigation, A.Č. and A.R.; resources, A.Č., A.R. and K.K.; data curation, A.Č. and A.R.; writing—original draft preparation, A.Č. and A.R.; writing—review and editing, A.Č., A.R. and K.K.; visualization, A.Č. and A.R.; supervision, A.Č., A.R. and K.K.; project administration, A.Č., A.R. and K.K.; funding acquisition, A.Č., A.R. and K.K. All authors have read and agreed to the published version of the manuscript.

**Funding:** This research received no external funding.

**Institutional Review Board Statement:** Not applicable.

**Informed Consent Statement:** Not applicable.

**Data Availability Statement:** Not applicable.

**Conflicts of Interest:** The authors declare no conflict of interest.

## Appendix A

**Table A1.** Environmental attitude measures.

| Measures Adapted from: | Construct | Items | Scale |
|---|---|---|---|
| Dunlap et al. [47] | Environmental attitude | We are approaching the limit of the number of people the Earth can support<br>Humans have the right to modify the natural environment to suit their needs<br>When humans interfere with nature, it often produces disastrous consequences<br>Humans are severely abusing the environment<br>The balance of nature is strong enough to cope with the impacts of modern industrial nations<br>The so-called "ecological crisis" facing humankind has been greatly exaggerated<br>The Earth is like a spaceship with very limited room and resources<br>Humans were meant to rule over the rest of nature<br>The balance of nature is very delicate and easily upset<br>Humans will eventually learn enough about how nature works to be able to control it<br>If things continue on their presen tcourse, we will soon experience a major ecological catastrophe | Strongly disagree (1)–strongly agree (7) |

**Table A2.** Environmental values measures.

| Measures Adapted from: | Construct | Items | Scale |
|---|---|---|---|
| Howell [87], Van Riper and Kyle [88] | Biospheric values | Protecting the environment (preserving nature)<br>Respecting the earth (harmony with other species)<br>Preventing pollution (protecting natural resources)<br>Unity with nature (fitting into nature) | Strongly disagree (1)–strongly agree (7) |
| | Altruistic values | Social justice (correcting injustice, care for the weak)<br>Equality (equal opportunity for all)<br>A world at peace (free of war and conflict) | |
| | Egoistic values | Influential (having an impact on people and events)<br>Wealth (material possessions, money)<br>Authority (the right to lead or command)<br>Social power (control over others, dominance)<br>Ambitious (hard-working, aspiring) | |

**Table A3.** Personal norms.

| Measures Adapted from: | Construct | Items | Scale |
|---|---|---|---|
| Vining and Ebreo [45] | Personal norms | I feel a strong personal obligation to recycle a large portion of my household recyclables<br>I am willing to go blocks out of my way to recycle household materials on a regular basis<br>For me, recycling is just a matter of money; I would not recycle material if I did not get paid back<br>I would recycle household materials whether or not I received payment<br>I would feel guilty if I did not recycle a large portion of my household recyclables | Strongly disagree (1)–strongly agree (7) |

**Table A4.** Perceived responsibility measures.

| Measures Adapted from: | Construct | Items | Scale |
|---|---|---|---|
| Paço and Rodrigues [46] | Perceived responsibility | I should be responsible for protecting our environment<br>Environmental protection starts with me<br>I think I have responsibility in protecting the environment in my country<br>I have taken responsibility for environmental protection since I was young<br>I am willing to take up responsibility to protect the environment in my country | Strongly disagree (1)–strongly agree (7) |

**Table A5.** Pro-environmental and prosocial engagement.

| Measures Adapted from: | Construct | Items | Scale |
|---|---|---|---|
| Kadic-Maglajlic et al. [66] | Pro-environmental and prosocial engagement | I like to learn more about [environmentally/socially]-friendly behavior<br>I keep up with things related to [environmentally/socially]-friendly behavior<br>Anything related to [environmentally/socially]-friendly behavior grabs my attention<br>I am heavily into [environmentally/socially]-friendly behavior<br>I am passionate about [environmentally/socially]-friendly behavior<br>My days would not be the same without [environmentally/socially]-friendly behavior<br>I enjoy [environmentally/socially]-friendly actions more when I am with others<br>[Environmentally/Socially]-friendly actions are more fun when other people around me do it too | Strongly disagree (1)–strongly agree (7) |

**Table A6.** Sustainable consumption behaviour.

| Measures Adapted from: | Construct | Items | Scale |
|---|---|---|---|
| Quoquab, Mohammad and Sukari [7] | Sustainable consumption behavior | I always try hard to reduce misuse of goods and services (e.g., I switch off the light and fan when I am not in the room<br>I recycle daily newspaper (e.g., use as pet's litter box, etc.)<br>I avoid being extravagant in my purchases<br>I reuse paper to write on the other side<br>While dining in restaurant, I order food(s) of only the amount that I can eat in order to avoid wasting food<br>I choose to buy product(s) with a biodegradable container or packaging when I am not in the room)<br>I do not like to waste food or beverages<br>I use eco-friendly products and services<br>I purchase and use products which are environmentally friendly<br>I often pay extra money to purchase environmentally friendly products (e.g., organic food)<br>I am concerned about the shortage of natural resources<br>I prefer to use a paper bag since it is biodegradable<br>I always remember that my excess consumption can create hindrance for the future generation to meet their basic needs<br>I care for the need fulfilment of the next generation<br>I often think about future generations' quality of life<br>I try to control my desire for excessive purchase for the sake of future generations<br>I am concerned about future generations<br>I try to minimise the excess consumption for the sake of preserving environmental resources for future generations | Strongly disagree (1)–strongly agree (7) |

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
