# Peer review of "Engaging in Sustainable Consumption: Exploring the Influence of Environmental Attitudes, Values, Personal Norms, and Perceived Responsibility"

_sustainability, doi:10.3390/su141610290_

Round 1

Reviewer 1 Report

This paper attempts to explore the associations among environmental attitude, environmental values, personal norms, perceived responsibility, pro-environmental and prosocial consumer engagement in sustainable consumption, and sustainable consumption behavior. The paper is well written. My concern on the paper is that Figure 2 and corresponding additional hypotheses can be integrated with Figure 1 and the hypotheses in Section 2 because these additional hypotheses are derived from the questionnaire survey results which use a three-dimension measurement of environmental values. These additional hypotheses are not developed theoretically. The authors can consider putting these additional hypotheses (Ha2, H2b, and H2c) in Section 2.2, and discuss their formulations theoretically.

Author Response

To: Reviewer

Re: The manuscript „ ENGAGING IN SUSTAINABLE CONSUMPTION: EXPLORING THE INFLUENCE OF ENVIRONMENTAL ATTITUDES, VALUES, PERSONAL NORMS, AND PERCEIVED RESPONSIBILITY” Authors: AistÄ— ÄŒapienÄ—, Aušra RÅ«telionÄ—, and Krysztof Krukowski 

Dear Reviewer,

Thank you for your letter accepting the manuscript entitled „ ENGAGING IN SUSTAINABLE CONSUMPTION: EXPLORING THE INFLUENCE OF ENVIRONMENTAL ATTITUDES, VALUES, PERSONAL NORMS, AND PERCEIVED RESPONSIBILITY” pending revisions. We have made all the changes you suggested in your letter.

Here detailed explanations:

Reviewers‘ comments 

Explanation 

The paper is well written. My concern on the paper is that Figure 2 and corresponding additional hypotheses can be integrated with Figure 1 and the hypotheses in Section 2 because these additional hypotheses are derived from the questionnaire survey results which use a three-dimension measurement of environmental values. These additional hypotheses are not developed theoretically. The authors can consider putting these additional hypotheses (Ha2, H2b, and H2c) in Section 2.2, and discuss their formulations theoretically. 

Thank you for the comments and suggestions. The changes were made. We joined two figures into one (see picture 1). The hypotheses (Ha2, H2b, and H2c) formulations discussed theoretically. (Lines:163-165; 184-186; 251-259) 

We believe the paper is now acceptable for publication and look forward to your response to the changes we have made.

Your sincerely,

Dr. AistÄ— ÄŒapienÄ—

Reviewer 2 Report

Sustainability ms 1805693

Engaging in sustainable consumption: Exploring the influence...

This is an interesting study, examining how people in Lithuania demonstrate that pro-environmental and pro-social consumer engagement mediate the relations between several antecedent factors (e.g., personal norms, environmental values) and self-reported sustainable consumption behaviors. The study features a large sample, the writing is good, and the analyses seem reasonably well conducted. Below are suggestions that I believe could improve this interesting paper.

1)  One of the primary contributions of the current work is to shed light on how people in Lithuania respond on these measures. However, the authors did not provide much detail about how well their participants "look like average Lithuanians." I found the age range of the current work to be especially impressive for an on-line sample, but could the authors speak more to the extent to which the sample is representative of Lithuanians in general (especially when the authors relied on social networking sites such as Facebook and Instagram to obtain participants)? The current work could more effectively make the case that it's providing insights into Lithuanians if it better establishes that its sample "looks like typical citizens" as well.

2)  I found the diversity in age, gender, education, and income in the current sample to be interesting (and I appreciate the authors providing the analyses in Table 4 to examine correlations among these demographic variables). These analyses led me to wonder about whether these factors might serve as moderators (e.g., might younger people show stronger effects than older people because "their future" is more at stake in sustainability)? Perhaps the authors could explore moderated mediation and add even deeper insights to their analyses. Relatedly, in the discussion (page 15), the authors attempted to explain a somewhat paradoxical finding that younger people have more pro-environmental attitudes but that older people engage in more sustainable consumption behaviors. One possibility here is that many "green behaviors" (e.g., more eco-friendly products, EV cars) can be more expense than less pro-environmental behaviors. Thus, old people have "have the means to be green" whereas younger people are more concerned about being green but don't have the income to do so (perhaps the SES data collected by the authors could speak to this possibility).

3)  I was a bit surprised to see that egoistic values had a positive (albeit small) relation to pro-environmental outcomes (e.g. responsibility, sustainable consumption behavior) when some research (e.g., McConnell & Jacobs; Schultz) find egoism predicts worse pro-environmentalism. Like most of the literature, the current work found that biospheric values were an especially strong predictor of sustainability (deGroot & Steg; Schultz), but the current findings led me to wonder whether egoistic values "look different" in Lithuania (e.g., perhaps the country is greater in collectivism or interdependent self-construals, which means "self interest" might not be so detrimental to nature).

4)  The mediation results (Tables 5-10) were a bit hard to follow, and I would have preferred the authors presented their path findings with a figure (or worst case, separate figures for each antecedent) rather than tables. Relatedly, because of the interrelations among the antecedent variables, I was curious about what the data might look like when entering them all simultaneously in mediational analyses (rather than piecemeal, one at a time)... perhaps some antecedents "matter more" than others and looking at the unique variance accounted for could shed light on this possibility.

5)  The authors discussed "emotional experiences" a couple of times in their introduction (e.g., page 2, 5) but they don't seem to assess it. There is a lot of literature on emotions and sustainability, especially on discrete positive or self-transcendent emotions (e.g., Jacobs & McConnell; Shiota et al.; Stellar et al.), but the authors discussion at present seems spotty and not especially germane to their current data collection. It seems the authors should either bolster this aspect of their work or cut it from the current write up.

6)  Minor point -- at times the authors use commas instead of period for decimal points. The authors should be consistent and follow journal style throughout.

Reviewer 3 Report

Well presented paper which confirms previous studies in this space. It would benefit from criticality in the discussion, for example, in relation to considering the finding that older consumers are more engaged - this could simply be because it is those older people who are more engaged who responded to the survey, and those older people who are less engaged chose not to take part in the survey. Such consideration re: the research methodology and method ought to be made clear. 
